# Supplementation with Two New Standardized Tea Extracts Prevents the Development of Hypertension in Mice with Metabolic Syndrome

**DOI:** 10.3390/antiox11081573

**Published:** 2022-08-15

**Authors:** Mario de la Fuente Muñoz, María de la Fuente Fernández, Marta Román-Carmena, Maria del Carmen Iglesias de la Cruz, Sara Amor, Patricia Martorell, María Enrique-López, Angel Luis García-Villalón, Antonio Manuel Inarejos-García, Miriam Granado

**Affiliations:** 1Physiology Department, Faculty of Medicine, Universidad Autónoma de Madrid, 28029 Madrid, Spain; 2R&D Department Biopolis, 46980 Paterna, Spain; 3R&D Department of Functional Extracts, ADM^®^, 46740 Valencia, Spain; 4CIBER Fisiopatología de la Obesidad y Nutrición, Instituto de Salud Carlos III, 28029 Madrid, Spain

**Keywords:** hypertension, obesity, metabolic syndrome, white tea, green tea, black tea, extract, antioxidant

## Abstract

Hypertension is considered to be both a cardiovascular disease and a risk factor for other cardiovascular diseases, such as coronary ischemia or stroke. In many cases, hypertension occurs in the context of metabolic syndrome (MetS), a condition in which other circumstances such as abdominal obesity, dyslipidemia, and insulin resistance are also present. The high incidence of MetS makes necessary the search for new strategies, ideally of natural origin and with fewer side effects than conventional pharmacological treatments. Among them, the tea plant is a good candidate, as it contains several bioactive compounds such as caffeine, volatile terpenes, organic acids, and polyphenols with positive biological effects. The aim of this study was to assess whether two new standardized tea extracts, one of white tea (WTE) and the other of black and green tea (CTE), exert beneficial effects on the cardiovascular alterations associated with MetS. For this purpose, male C57/BL6J mice were fed a standard diet (Controls), a diet high in fats and sugars (HFHS), HFHS supplemented with 1.6% WTE, or HFHS supplemented with 1.6% CTE for 20 weeks. The chromatography results showed that CTE is more concentrated on gallic acid, xanthines and flavan-3-ols than WTE. In vivo, supplementation with WTE and CTE prevented the development of MetS-associated hypertension through improved endothelial function. This improvement was associated with a lower expression of proinflammatory and prooxidant markers, and—in the case of CTE supplementation—also with a higher expression of antioxidant enzymes in arterial tissue. In conclusion, supplementation with WTE and CTE prevents the development of hypertension in obese mice; as such, they could be an interesting strategy to prevent the cardiovascular disorders associated with MetS.

## 1. Introduction

According to data from the World Health Organization (WHO), cardiovascular diseases are the leading cause of death in the world, causing around 18 million deaths annually [1]. Among them, systemic arterial hypertension (SAH) is the main risk factor for premature death worldwide, with upwards of one in four men and one in five women suffering from this condition [1]. SAH is considered both a disease and a risk factor for the development of other cardiovascular diseases such as coronary heart disease, stroke, heart failure, atrial fibrillation, and chronic kidney disease, among others [2]. The most prevalent type of SAH is essential arterial hypertension (EAH), which is a very heterogeneous polygenically based disorder that is also influenced by acquired or environmental factors. These environmental factors include increased salt and/or decreased potassium and calcium consumption in the diet, excessive alcohol intake, and a sedentary lifestyle [3]. However, one of the main risk factors for suffering EAH is overweightness and obesity [3,4]. In fact, excess weight gain, especially when associated with increased visceral adiposity, is the major cause of hypertension, accounting for 65–75% of the risk for EAH [4].

Both obesity and hypertension are usually present in a more complex disorder called metabolic syndrome (MetS). MetS is defined as the association of at least three of the following cardiometabolic alterations in the same individual: visceral obesity, hypertension, insulin resistance, hyperglycemia and dyslipidemia (increased circulating levels of LDL-cholesterol and/or decreased concentrations of HDL-cholesterol). The incidence of MetS has dramatically increased in the last few decades, not only in developed countries but also in developing ones, constituting a huge sanitary and economic challenge worldwide [5,6] due to its associated comorbidities, especially cardiovascular diseases and type II diabetes [7].

In MetS, insulin resistance seems to play a key role in the development of metabolic and cardiovascular alterations, including hypertension [8]. Insulin resistance contributes to the development of hypertension through different mechanisms, which include endothelial dysfunction [9], increased sympathetic nervous system activity [10], and enhanced sensitivity to components of the renin-angiotensin-aldosterone system [11].

Among the physio-pathological mechanisms that are involved in the development of cardiovascular alterations in MetS are the low grade of chronic inflammation and increased oxidative stress [12]. Both phenomena are key in the development of endothelial dysfunction, which results in an unbalance in the production/release of vasodilators and vasoconstrictors by the vascular endothelium [13]. In addition, inflammation and increased oxidative stress are involved in the development of hypertension through the hyperactivity of the sympathetic nervous system [14,15], as well as by affecting the sensitivity of vascular smooth muscle cells to vasodilators such as NO or insulin and vasoconstrictors such as angiotensin II [16] or endothelin-1 [17]. For this reason, anti-inflammatories, and especially antioxidants, have been proposed by some authors as potential agents to treat and/or prevent the development of hypertension [18,19,20].

Among these agents is tea, a beverage made from the leaves of *Camellia sinensis;* it originated in China, and has been used for medicinal purposes from ancient times. The tea plant is consumed in over 160 countries, and is cultivated in more than 30. There are different varieties of tea depending on the grade of the leaves’ fermentation. According to this, tea can be classified into unfermented tea (yellow, white and green), partially fermented tea (pu-erh and oolong), and fully fermented tea (black). All of the varieties contain different compounds such as caffeine, volatile terpenes, organic acids, and polyphenols that provide each type of tea with is characteristic flavor and taste [21]. Phenolic compounds, mainly catechins, tannins and their derivatives, are responsible for the antioxidant properties, and have a strong impact on the color and aroma of tea infusions, determining their characteristic bitter and astringent flavor. The content in phenolic compounds varies among the different tea types, and depends on the technological process used. Flavan-3-ols are the most abundant phenolic compounds in fresh tea leaves [22], whereas monomeric catechins such as epigallocatechin, epicatechin, epicatechin gallate and epigallocatechin gallate predominate in green tea [23]. During fermentation from green to oolong tea, catechins are partially oxidized to dimers like theaflavins and other polyphenols of high molecular weight. Finally, in fully fermented black tea, most catechins are transformed into thearubigins and theaflavins [24].

Because the metabolic and cardiovascular alterations associated with MetS are due, at least in part, to increased oxidative stress, tea supplementation has been suggested as a possible strategy to ameliorate these alterations. Indeed, there are several epidemiological studies that report that the consumption of tea infusions, specially green tea, has a beneficial effect, lowering body weight, adiposity and the risk of suffering type II diabetes [25]. Likewise, tea ingestion is reported to decrease mortality due to cardiovascular diseases by reducing hypertension and LDL oxidation, thus reducing the incidences of coronary heart disease and stroke [25]. Unfortunately, these beneficial properties of tea may be affected by adulteration and falsification processes. Indeed, some commercial tea products present considerable amounts of unexpected compounds that are not characteristic of *Camellia sinensis’* profile, such as synthetic antioxidants, organic solvents, and intermediate products of catechin synthesis [26]. Thus, it is mandatory to demonstrate the volatile profile of tea extracts in order to guarantee the authenticity, safety and the quality of the product. For this reason, the aim of this study was to test the possible beneficial effects of two standardized tea extracts, one of white tea extract and the other a proprietary blend of green and black teas (Complex Tea Extract) in the cardiovascular alterations associated to MetS in mice.

## 2. Materials and Methods

### 2.1. Reagents and Chemicals

The following standards were used: xanthines (caffeine, theobromine and theophylline), monomeric flavan-3-ols ((+)-catechin, catechin gallate, (−)-Epicatechin, Epicatechin-3-gallate, epigallocatechin, epigallocatechin-3-gallate (EGCg), green tea catechin mix), theaflavins (theaflavin, theaflavins mix (tea extract from Camellia sinensis)) and gallic acid. They were purchased from Merck (Darmstadt, Germany) and Phytolab (Vestenbergsgreuth, Germany) for the identification and/or quantification of characteristic bioactive components from tea *(Camellia sinensis)*. Trifluoroacetic acid, acetic acid, acetonitrile, dimethyl sulfoxide, and water of chromatographic quality were purchased from VWR (Barcelona, Spain).

#### Commercial Tea Extracts

Two different commercial tea powdered extracts were employed in the present study: (1) a proprietary blend of green and black tea leaves named ADM^®^ Complex Tea Extract (CTE), standardized to total flavan-3-ols (monomeric and theaflavins) and methylxanthines (caffeine, theobromine and theophylline), and (2) ADM^®^ White Tea extract (WTE), a white tea extract standardized to monomeric flavan-3-ols and xanthines.

### 2.2. Standards Preparation

Four working calibration standards solutions (EGCg, caffeine, theaflavin and gallic acid) were employed for the quantification of the main groups of components in the tea functional powdered extracts.

### 2.3. High Performance Liquid Chromatography (HPLC)

Flavan-3-ols, xanthines, theaflavins and gallic acid analyses were performed according to Lee and Ong et al. [27]. The HPLC equipment used for the analysis consisted of a Shimadzu NEXERA XR UHPLC 70MPa coupled to a photodiode array detector SPD-M40 model (Izasa Scientific, Madrid, Spain). The chromatographic analyses were performed by an octadecyl silane column ZORBAX ECLIPSE PLUS C18 (250 mm, 4.6 mm, 5 µ) together with its corresponding precolumn (Agilent Technologies, Barcelona, Spain).

Detection was performed at 275nm, the temperature of the oven was set at 32ºC, the work flow was maintained at 1.0 mL/min, and the injection volume was 2 µL. The binary gradient system used for the chromatographic separation consisted of Phase (A) 5% (*v*/*v*) acetonitrile 0.035 (*v*/*v*) trifluoroacetic acid and Phase (B) 50% (*v*/*v*) acetonitrile 0.025% (*v*/*v*) trifluoroacetic acid. The initial conditions were set with A-B (90:10), and the gradient was slightly increased to 20% at 10 min, and to 40% from 25 to 27 min. Finally, the column was again balanced to the initial gradient conditions for 3 min before the next injection.

The identification of monomeric and oligomeric flavan-3-ols was performed by comparing the retention time and UV-Vis spectra of the corresponding standards. Quantification was performed using an external calibration curve with at least five different concentration points (r^2^ = 0.99); the results are expressed in percentages (%, dry basis). The sum of monomeric flavan-3-ols, xanthines and theaflavins were respectively quantified as EGCg and theaflavin equivalents.

### 2.4. Animals

For the in vivo study, all of the experiments were conducted according to the European Union Legislation and with the approval of the Animal Care and Ethical Committee of the Community of Madrid (Madrid, Spain) (PROEX 214.1_20). 

Forty 16-week-old C57/BL6J mice were housed two per cage and maintained in climate-controlled quarters under controlled conditions of humidity (50–60%) and temperature (22–24 °C), and with a 12 h light cycle. The mice were fed ad libitum, and were divided into four experimental groups: mice fed with a standard chow (Chow; n = 10); mice fed a high fat/high sucrose (HFHS) diet containing 58% kcal from fat with sucrose (HFHS; n = 10), mice fed a high fat/high sucrose (HFHS) diet containing 58% kcal from fat with sucrose supplemented with 1.6% White Tea Extract (HFHS + WTE; n = 10), and mice fed a high fat/high sucrose (HFHS) diet containing 58% kcal from fat with sucrose supplemented with 1.6% Complex Tea Extract (HFHS + CTE; n = 7). The customized diets were elaborated by the company Research Diets Inc. (New Brunswick, NJ, USA). The tea extracts were added to the commercial high fat/ high sucrose diet with reference D12331 (https://researchdiets.com/formulas/d12331, accessed date 1st July 2022). The caloric content of the tea extracts was taken into consideration in order for the diets to be isocaloric. The diet with reference 11112201 (D11112201) was used as the standard diet (chow). 

The dosage of tea extracts was chosen according to previous studies in which doses between 0.5 and 5% were shown to exert beneficial effects on body weight reduction and metabolism in rodents with MetS [28,29,30]. Studies in humans used dosages between 10 and 20 times higher than the one used in this study [31,32], which is in the accepted range when dosages for rodents are converted into dosages for humans [33].

All of the mice were maintained on their diets for 20 weeks. A weekly control of solid and liquid intake and body weight was performed. After the 20 weeks of diet consumption, all of the animals were injected with an overdose of sodium pentobarbital (100 mg/kg) and killed by decapitation after overnight fasting.

### 2.5. Measurement of the Mean Arterial Pressure (MAP) in Conscious Mice by the Tail-Cuff System

The mean arterial blood pressure (MBP) was measured by tail-cuff plethysmography using a Niprem 645 blood pressure system (Cibertec, Madrid, Spain during the four weeks before sacrifice. The measurements were performed in each mouse three times/week. For that purpose, the mice were first habituated to the experimental conditions for at least 3 days. Prior to the MBP measurements, the mice were prewarmed to 34 °C for 10–15 min. Then, an occlusion cuff and a sensor were placed at the base of the tail. The occlusion cuff was inflated to 250 mm Hg and deflated over 20 s. Each day, three to four measurements were recorded in each mouse, and the average was calculated for each animal.

### 2.6. Experiments of Heart Perfusion: Langendorff

The hearts were immediately mounted in the perfusion system (Langendorff). The left ventricular pressure was measured using a latex balloon inflated to a diastolic pressure of 5–10 mmHg, and the coronary perfusion pressure was measured through a lateral connection in the perfusion cannula trough Statham transducers (Statham Instruments, Los Angeles, CA, USA). The left ventricular pressure was recorded, and was used to calculate the first derivative of the left ventricular pressure curve (dP/dt) as an index of heart contractility and the heart rate. After a 30 min equilibration period with constant flow perfusion, global ischemia was induced by stopping the flow perfusion for 30 min. Afterwards, the hearts were re-perfused for 45 min. After ischemia-reperfusion (IR), the hearts were collected and stored at −80 °C for further analysis. Data were recorded using the PowerLab/8e data acquisition system (ADInstruments, Colorado Springs, CO, USA).

### 2.7. Experiments of Vascular Reactivity

Next, 2-mm aorta segments were prepared and kept in cold isotonic saline solution. Abdominal segments were used for the vasoconstriction studies, and thoracic segments were used for the vasodilation studies. The segments were placed in a 4-mL organ bath for the recording of the isometric tension using a PowerLab data acquisition system (ADInstruments, Colorado Springs, CO, USA). After equilibration for 60–90 min, an optimal passive tension of 1 g was applied. Afterwards, potassium chloride (100 mM, Merck Millipore, Burlington, MA, USA) was added to the organ bath in order to determine the smooth muscle contractility. Contractions below 0.5g in response to KCl were discarded. 

Dose–response curves were created in response to the vasoconstrictors norepinephrine (10^−9^–10^−4^ M), angiotensin-II (10^−11^–10^−6^ M) and endothelin-1 (10^−10^–10^−6.5^ M) (Sigma-Aldrich, St. Louis, MO, USA). The contraction in response to each dose of the vasoconstrictors was represented as the % contraction to KCl.

For the relaxation experiments, prior to the dose-response curves in response to sodium nitroprusside (NTP; 10−9–10−5 M) and acetylcholine (10−9–10−4 M), the segments were precontracted with U46619 10^−7.5^M (Sigma-Aldrich, St. Louis, MO, USA). The effect of oxidative stress on endothelial function was assessed by preincubating some of the thoracic segments with apocynin 10^−6^ M for 30 min before the ACh dose-response curves. The % of relaxation in response to each ACh dose was calculated by referring to the maximum relaxation (Emax) in response to a dose of NTP 10^−5^ M.

### 2.8. RNA Preparation and Quantitative Real-Time PCR

The total RNA was extracted from aortic tissue using the Tri-Reagent protocol. cDNA was then synthesized from 1 µg total RNA using a high-capacity cDNA RT kit (Applied Biosystems, Foster City, CA, USA). The mRNA levels were assessed by using assay on-demand kits (Applied Biosystems) for each gene. TaqMan Universal PCR Master Mix (Applied Biosystems, Foster City, CA, USA) was used for amplification according to the manufacturer’s protocol in a Step One System (Applied Biosystems, Foster City, CA, USA).

The mRNA concentrations of interleukin-6 (IL-6) (Mm00446190_m1), interleukin-1 beta (IL-1β) (Mm00434228_m1), tumor necrosis factor-alpha (TNF-α) (Mm00443258_m1), monocyte chemoattractant protein (MCP-1) (Mm00441242_m1), glutathione reductase (GSR) (Mm00439154_m1), NADPH oxidase-4 (NOX-4) (Mm00479246_m1), superoxide dismutase 1 (SOD-1) (Mm01344233_g1), Glutathione Peroxidase 3 (GPX3) (Mm00492427_m1), endothelial nitric oxide synthase (eNOS) (Mm00435217_m1), and alpha-1 adrenergic receptor (α1 adre.) (Mm00442668_m1) were assessed by quantitative real-time PCR that was performed using assay on-demand kits (Applied Biosystems) for each gene. TaqMan Universal PCR Master Mix (Applied Biosystems, Foster City, CA, USA) was used for amplification according to the manufacturer’s protocol in a Step One System (Applied Biosystems, Foster City, CA, USA). The values were normalized to the housekeeping Hypoxanthine Phosphoribosyltransferase 1 (HPRT1) (Mm03024075_m1). In order to determine the relative expression levels, the ΔΔCT method was used, and all of the data are expressed as a percentage of the control group (Chow).

### 2.9. Vascular superoxide anion production

The vascular production of superoxide anions was assessed using the oxidative fluorescent dye dihydroethidium (DHE, Invitrogen Life Technologies, Carlsbad, CA, USA; Cat-No. D23107). Briefly, arterial sections were equilibrated at 37 °C for 30 min in Krebs-HEPES buffer. DHE dissolved in fresh buffer (2 µmol/L) was added to each tissue section, cover-slipped, and incubated in a dark and humidified chamber for 30 min at 37 °C. Images were obtained using a confocal microscope (Leica TCS SP2 equipped with a krypton/argon laser, ×40 objective; Leica Microsystems). Fluorescence was detected with a 568-nm long-pass filter. The same imaging setting was used for all of the experimental conditions. For quantification, the mean fluorescence density in the target region was analyzed; two to three rings per animal from each experimental group were sampled and averaged. All of the samples were analyzed in the same day. The data were expressed as a percentage of the signal in the control arteries.

### 2.10. Statistical Analysis

One-way ANOVA followed by a Bonferroni post-hoc test was used for the statistical data analysis using GraphPad Prism 8.0 (San Diego, CA, USA). All of the values are expressed as means ± the standard error of the mean (SEM). A *p*-value of ≤ 0.05 was considered statistically significant.

## 3. Results

### 3.1. Chemical Characterization of Tea Extracts by HPLC

As shown in Figure 1, CTE and WTE showed quantitative and qualitative differences regarding the chromatographic composition. Compared to WTE, the CTE is more concentrated on gallic acid (0.4–0.8%, dry basis), xanthines (4.7–6.5%, dry basis) and flavan-3-ols (8.9–12.4%, dry basis). Flavan-3-ols from CTE are composed of monomeric and oligomeric flavan-3-ols (Theaflavins), whereas WTE is just composed of monomeric Flavan-3-ols (Figure 1).

### 3.2. Results from the In Vivo Experiment

#### 3.2.1. Body Weight, Glycaemia and Lipid Profile

The consumption of the HFHS diet induced a signficant increase in body weight, glycaemia, and the plasma levels of triglycerides, total cholesterol, LDL-c and HDL-c (Table 1; *p* < 0.001 for all). Moreover, the triglycerides and total cholesterol levels were significantly higher in obese mice supplemented with WTE compared to HFHS mice (*p* < 0.05 for both). Neither supplementation with WTE nor with CTE attenuated the obesity-induced changes in glycaemia or the lipid profile. However, supplementation with CTE induced a significant reduction in body weight (*p* < 0.05).

#### 3.2.2. Effects of CTE and WTE on Cardiac Function

Changes in heart contractility (dp/dt), coronary pressure, and heart rate were assessed in the hearts of mice from the different experimental groups before and after being subjected to ischemia reperfusion (IR).

Before IR, no significant changes in cardiac function were found among the experimental groups (Figure 2A). IR did not modify coronary pressure, but it significantly decreased both heart rate (F = 9.91; *p* < 0.01) and dp/dt (F = 19.45; *p* < 0.001) equally in all of the groups.

#### 3.2.3. Effects of CTE and WTE on the Gene Expression of Inflammatory and Oxidative Stress Related Markers in Cardiac Tissue

Figure 3 shows the gene expression of inflammatory and oxidative stress related markers in cardiac tissue after IR.

Neither the consumption of the HFHS diet nor the supplementation with the tea extracts modified the gene expression of IL-1β, IL-6, TNF-α, GPX-3, GSR and SOD-1 in cardiac tissue among the experimental groups.

### 3.3. Effects of CTE and WTE on Blood Pressure and Vascular Reactivity

After 20 weeks of consumption of the HFHS diet, obese mice showed a significant increase in blood pressure (BP) compared to mice fed with chow (Figure 4; *p* < 0.001). Supplementation with both WTE and TC significantly attenuated the obesity-induced increase in arterial mean pressure (*p* < 0.01 and *p* < 0.001, respectively).

In order to elucidate the mechanisms involved in the antihypertensive effects of the tea extracts, experiments of vascular reactivity were performed in aorta segments from mice of the different experimental groups.

The curve in response to NTP showed that the endothelium independent relaxation was unchanged among the experimental groups (Figure 5A). However, the endothelium dependent relaxation represented by the response of aorta segments to ACh was significantly reduced in the aorta segments from obese mice compared to the controls at all the studied doses (Figure 5B; *p* < 0.05). This reduced response of aorta segments to ACh was associated with a significant decrease in the gene expression of the endothelial nitric oxide synthase enzyme (eNOS) (Figure 5D; *p* < 0.05), and was prevented by the pre-incubation of the segments with the antioxidant apocynin (Figure 3C; *p* < 0.05).

Supplementation with both WT and CTE prevented the reduced relaxation to Ach induced by obesity (*p* < 0.05 for both) and, in the case of CTE, it significantly increased the mRNA levels of eNOS (*p* < 0.05). Moreover, the arterial gene expression of eNOS was not different between obese mice supplemented with WTE and control mice. Finally, despite the untreated obese mice, supplementation with WTE or CTE did not affect the relaxation of the aorta segments in response to ACh after pre-incubation with apocynin.

In the vasoconstriction experiments, no significant changes were found among the experimental groups in the response of the aorta segments to AngII (Figure 6A) or ET-1 (Figure 6B). However, the vascular response of the aorta segments to NA was significantly increased in the aorta segments from obese mice supplemented with WT (*p* < 0.05). The changes in the vascular response of the aorta segments to NA were associated with a significant decreased in the gene expression of the α − 1 adrenergic receptor in aortic tissue from untreated obese mice (Figure 6D; *p* < 0.01) that was attenuated by the supplementation with both CTE and with WTE (*p* < 0.05 for both).

### 3.4. Effects of CTE and WTE on the Gene Expression of Inflammatory and Oxidative Stress-Related Markers in Arterial Tissue

The gene expression of pro-inflammatory and oxidative stress markers in arterial tissue is shown in Figure 7 and Figure 8, respectively.

Obesity was associated with an upregulation in the gene expression of IL-1β (*p* < 0.05), IL-6 (*p* < 0.01) and NOX-4 (*p* < 0.05), and with a downregulation in the mRNA levels of the antioxidant enzymes GPX-3 and SOD-1 (*p* < 0.01 for both) in aortic tissue. Supplementation with WTE prevented the obesity-induced changes in the gene expression of IL-1β, IL-6 and NOX-4 (*p* < 0.05 for all). Moreover, supplementation with CTE not only decreased the mRNA levels of IL-1β (*p* < 0.001), IL-6 (*p* < 0.01) and NOX-4 (*p* < 0.05) in arterial tissue, but also increased the gene expression of the antioxidant enzymes GPX-3 and SOD-1 (*p* < 0.01 for both).

Likewise, supplementation with WFTE and CTE reduced the obesity-induced increase in the content of superoxide anions in the aorta sections (Figure 9; *p* < 0.05 for both). Indeed, the content of this free radical was significantly lower in aorta sections from mice supplemented with the extracts than in the control animals (*p* < 0.05 for both).

## 4. Discussion

In this paper we describe the positive effects of two objectively standardized novel tea extracts on cardiovascular function.

The major components for both tea extracts are flavan-3-ols, with those present in WTE being exclusively monomers, characteristic of white tea. However, CTE also showed oligomeric flavan-3-ols in its composition, which are characteristic of black tea. Both tea extracts showed the typical chromatographic profile of *Camellia sinensis* (L.) *Kuntze,* with a complex composition of monomeric flavan-3-ols (n = 9 for both tea extracts), oligomeric flavan-3-ols (n = 4, only for CTE), methylxanthines (n = 3 for both tea extracts), and gallic acid. The chromatographic profile and characteristic proportions of molecules may help to avoid fraud, adulteration, and non-rational proportions of some components such as (−)-epigallocatechin-3-gallate (EGCG), with concentration over 90% in most of commercial extracts [26]. This has recently activated all of the alarms of the EFSA organization, publishing the corresponding scientific opinion on safety due to the hepatotoxic effects of EGCG in amounts higher than 800 mg/day [34]. This regulation will help to limit the presence of tea extracts in the market with non-rational compositions and extreme concentrations of concrete flavan-3-ols such as EGCG, which may cause public health issues related to safety and toxicity.

In order to assess the possible beneficial effects of both tea extracts on the cardiovascular alterations associated to metabolic syndrome, we performed an in vivo model of metabolic syndrome in mice, and studied both cardiac and vascular function.

As expected, the consumption of the HFHS diet increased body weight, glycaemia and the circulating levels of triglycerides, total cholesterol, LDL-c and HDL-c. Neither supplementation with WTE nor with CTE had a positive impact reducing these alterations, except for body weight, which was significantly decreased in obese mice supplemented with CTE. The weight reduction effect of green tea has already been reported both in experimental animals [35] and in humans [31] with MetS, and seems to be due to both decreased lipid and protein absorption in the intestine, and the activation of the AMP- kinase pathway in the liver, skeletal muscle, and adipose tissues [36].

No significant changes were found in cardiac function, either in basal conditions or in response to IR among the experimental groups. Likewise, no differences were found among the experimental groups in the gene expression of different markers related to both inflammation and oxidative stress after IR in myocardial tissue.

The results for cardiac function disagree with previous studies in which epigallocatechin-3-gallate (EGCG), the main catechin present in green tea, induced electrocardiographic changes like the prolongation of QRS and PR intervals, the altered shape of the ST-T-wave segment, and the shortening of the QT interval [37]. Moreover, EGCG at nanomolar concentrations is reported to exert direct effects on cardiomyocytes, enhancing contractility by increasing electrically evoked Ca^2+^ transients, the ryanodine receptor type 2 (RyR2) channel open probability, and the sarcoplasmic reticulum (SR) Ca^2+^ content [38].

There are also several studies in experimental animals demonstrating that EGCG administration prevents the reduction in myocardial contractility induced by IR in rats [39,40,41,42], rabbits [43] and guinea pigs [44]. Likewise, pretreatment with a green tea extract administered before regional myocardial ischemia-reperfusion improved myocardial contractility through the attenuation of calcium overload in rats [45]. The beneficial effects of tea on cardiac function are the result, at least in part, of the presence of polyphenols which are reported to exert antioxidant effects in the heart both in vivo [46,47] and in vitro [48,49,50]. However, in this study we did not find differences in the gene expression of antioxidant enzymes among the experimental groups. The disagreement between our results and the previous studies may be explained, at least in part, by differences in species (mice vs. rats, rabbits or guinea pigs) and/or by differences in the extract compositions, dosages, or duration of the treatment. In addition, most of the experiments that have assessed the effects of EGCG or tea extracts on heart function are performed in control hearts, whereas in this study we have used hearts from mice with metabolic syndrome, a condition that may significantly affect the response of the myocardium to these insults. Therefore, further investigations are required in order to elucidate whether the effects of supplementation with CTE and WTE are different in normal conditions or within a context of metabolic syndrome.

The most important finding of this work is that supplementation with both WTE and CTE prevent the development of hypertension induced by the consumption of a high fat/high sucrose diet for 6 months in mice. This result is in agreement with previous studies that have reported that supplementation with green tea exerts antihypertensive effects in different animal models that include AngII-induced hypertension [51], spontaneous hypertensive rats [40], and hypertension associated to metabolic syndrome [52]. Likewise, in humans, the consumption of green tea is reported to have positive effects lowering systolic and diastolic blood pressure both in healthy subjects [53,54,55] and in patients with type-2 diabetes [56] and obesity [57,58]. Moreover, green tea consumption has been inversely associated with 5-year blood pressure values in the Chinese population [59]. Thus, it seems clear that green tea has a positive effect lowering blood pressure. However, to our knowledge, this is the first study reporting the beneficial effect of white tea consumption on cardiovascular function, and particularly on hypertension associated to metabolic syndrome.

The results from the vascular reactivity experiments reveal that the hypotensive effects of both tea extracts in obese mice are due to an improvement in endothelial function, and that this improvement is mediated, at least in part, through decreased inflammation and oxidative stress. Specifically, the supplementation of obese mice with both tea extracts significantly decreases the mRNA levels of the proinflammatory cytokines IL-6 and IL-1β, the gene expression of the prooxidant enzyme NOX-4, and the content of superoxide anions in arterial tissue. Moreover, supplementation with CTE prevented the obesity-induced decrease in the gene expression of the antioxidant enzymes GPX-3, GSR and SOD-1, and increased the mRNA levels of endothelial nitric oxide synthase (eNOS) in arterial tissue. These results agree with a previous study in which supplementation with a decaffeinated green tea extract to rats with MetS for 12 weeks improved endothelial dysfunction by reducing NADPH oxidase activity and the formation of ROS, and by activating the phosphorylation of Akt and eNOS in aortic tissue [52]. Likewise, the chronic consumption of black tea is reported to improve endothelial function in ovariectomized rats through the decreased production of ROS, which results in the increased phosphorylation of eNOS and increased NO bioavailability [60]. Again, no studies have so far evaluated the effects of supplementation with a white tea extract on endothelial function.

The beneficial effects on blood pressure are attributed to the different compounds present in the tea extracts such as EGCG [40,61] or L-theanine [62], which induce the phosphorylation of eNOS and the subsequent release of NO. This effect is due in large part to their antioxidant capacity. However, an important finding of this work is that the tea extracts tested in this study not only decrease oxidative stress but also inflammation in arterial tissue, which is an issue that may contribute to their antihypertensive effects. In this regard, and contrary to the antioxidant effect that is similar after the supplementation with both extracts, our results show that CTE exerts a more powerful anti-inflammatory effect in arterial tissue than WTE. This higher anti-inflammatory effect may be due, at least in part, to the different compositions of the extracts, and particularly to the higher amount of gallic acid, xanthines and flavan-3-ols in CTE compared to WTE. Indeed, the anti-inflammatory effects of gallic acid [63], xanthines [64] and flavan-3-ols [65] have been reported extensively, and may contribute to the more pronounced antihypertensive effect of CTE compared to WTE.

Finally, the supplementation with both tea extracts did not modify the response of aorta segments to the vasoconstrictors ET-1 and AngII, but it prevented the obesity-reduced gene expression of the α1-adrenergic receptor in aortic tissue. This result may indicate a positive effect of these extracts in attenuating the sympathetic hyperactivation associated with obesity, a condition that plays an important role in the development of hypertension in this condition [66].

## 5. Conclusions

In conclusion, supplementation with two new standardized tea extracts, one of white tea (WTE) and another of green and black tea (CTE), attenuates endothelial dysfunction and prevents the development of hypertension in mice with metabolic syndrome by decreasing arterial inflammation and oxidative stress. 

## Figures and Tables

**Figure 1 antioxidants-11-01573-f001:**
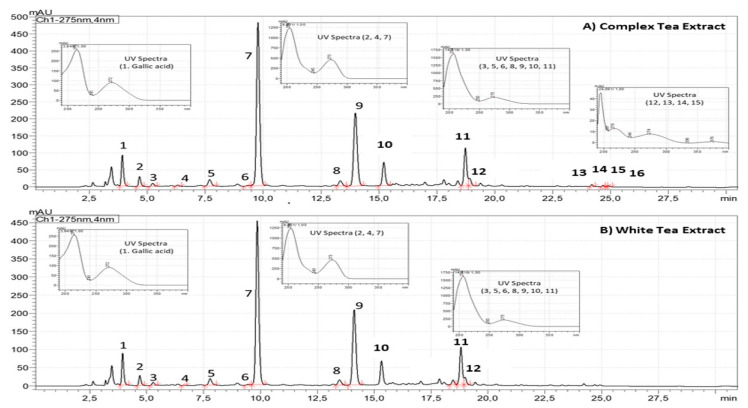
Chromatographic profiles of ADM^®^ Complex Tea (**A**) and White Tea (**B**) Powdered Extracts standardized to flavan-3-ols (3. Gallocatechin, 5. Epigallocatechin, 6. Catechin, 8. Epicatechin, 9. Epigallocatechin gallate, 10. Gallocatechin-3-gallate, 11. Epicatechin-3-gallate, 12. Catechin-3-gallate, 13. Theaflavin, 14. Theaflavin-3-monogallate, 15. Theaflavin-3’-monogallate, 16. Theaflavin-3,3´-gallate) and xanthines (2. Theobromine, 4. Theophylline and 7. Caffeine) analyzed by high-performance liquid chromatography coupled to photo diode array at 275 nm.

**Figure 2 antioxidants-11-01573-f002:**
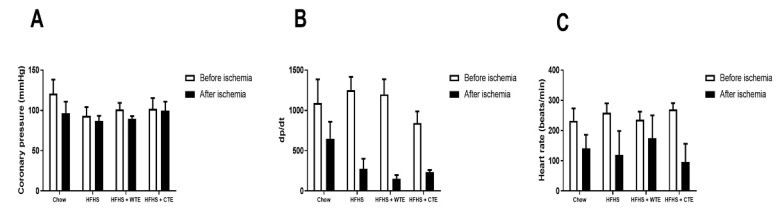
Changes in coronary perfusion pressure (**A**), contractility (**B**) and heart rate (**C**) before and after coronary ischemia-reperfusion in hearts from mice fed a standard diet (Chow), a high-fat diet/sucrose diet (HFHS), a high-fat diet/sucrose diet supplemented with White Tea Extract (HFHS + WTE), or a high-fat diet/sucrose diet supplemented with Tea Complex Extract (HFHS + TCE).

**Figure 3 antioxidants-11-01573-f003:**
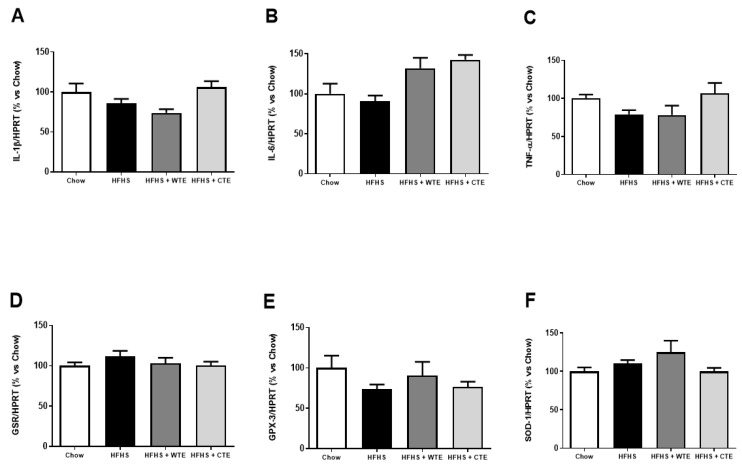
Gene expression of interleukin-1 beta (IL-1β) (**A**), interleukin-6 (IL-6) (**B**), tumor necrosis factor α (TNF-α) (**C**), glutathione reductase (GSR) (**D**), glutathione peroxidase 3 (GPX_3_) (**E**) and superoxide dismutase-1 (SOD-1) (**F**) in heart tissue after coronary ischemia-reperfusion from mice fed a standard diet (Chow), a high-fat diet/sucrose diet (HFHS), a high-fat diet/sucrose diet supplemented with White Tea Extract (HFHS + WTE), or high-fat diet/sucrose diet supplemented with Tea Complex Extract (HFHS + TCE). The values are represented as the mean ± S.E.M (n = 7–10 samples/experimental group), and expressed as a percentage vs. chow.

**Figure 4 antioxidants-11-01573-f004:**
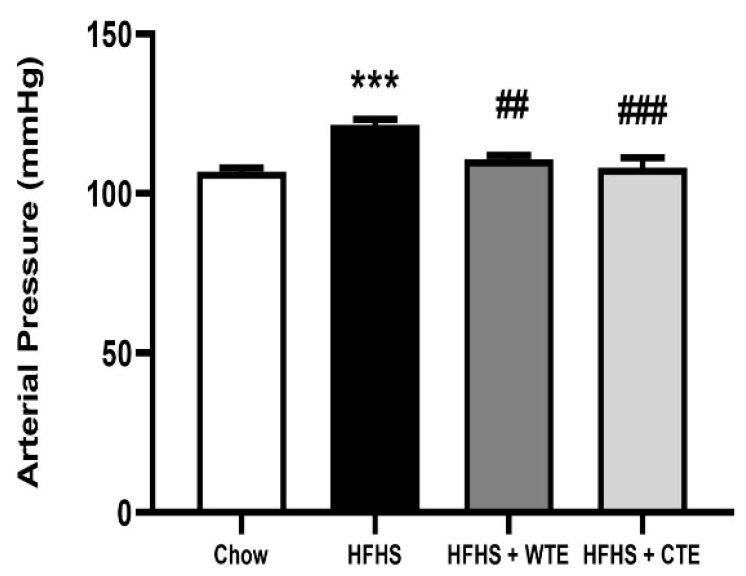
Mean arterial pressure of mice fed a standard diet (Chow), a high-fat diet/sucrose diet (HFHS), or high-fat diet/sucrose diet supplemented with White Tea Extract (HFHS + WTE), or high-fat diet/sucrose diet supplemented with Tea Complex Extract (HFHS + TCE). *** *p* < 0.001 vs. chow. ## *p* < 0.01; ### *p* < 0.001 vs. HFHS. The values are represented as the mean ± S.E.M; n = 7–10 samples/experimental group.

**Figure 5 antioxidants-11-01573-f005:**
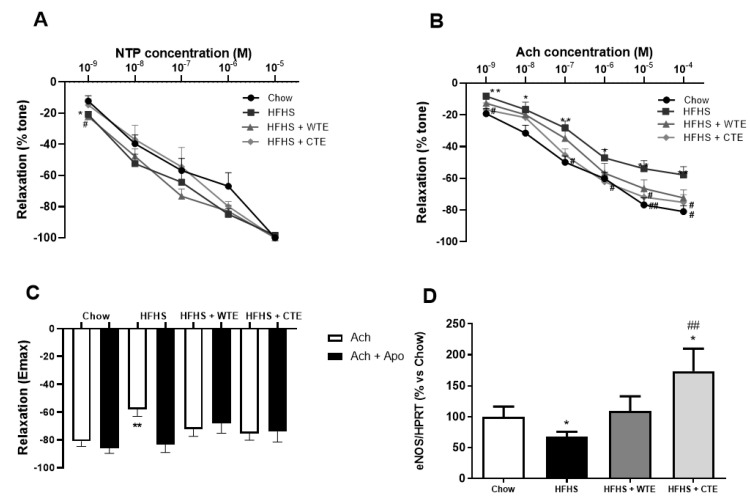
Relaxation of thoracic aortic segments to sodium nitroprusside (NTP) (10^−9^–10^−5^ M) (**A**), relaxation to acetylcholine (ACh) (10^−9^–10^−4^ M) (**B**), relaxation to acetylcholine in the presence/absence of apocynin (ACh/ACh + Apo) (10^−6^ M) (**C**), and the gene expression of endothelial nitric oxide synthase (eNOS) (**D**) of mice fed a standard diet (Chow), a high-fat diet/sucrose diet (HFHS), a high-fat diet/sucrose diet supplemented with White Tea Extract (HFHS + WTE), or a high-fat diet/sucrose diet supplemented with Tea Complex Extract (HFHS + TCE). * *p* < 0.05; ** *p* < 0.01 vs. chow. # *p* < 0.05 vs. HFHS; ## *p* < 0.01 vs. HFHS. The values are represented as the mean ± S.E.M; n = 7–10 samples/experimental group.

**Figure 6 antioxidants-11-01573-f006:**
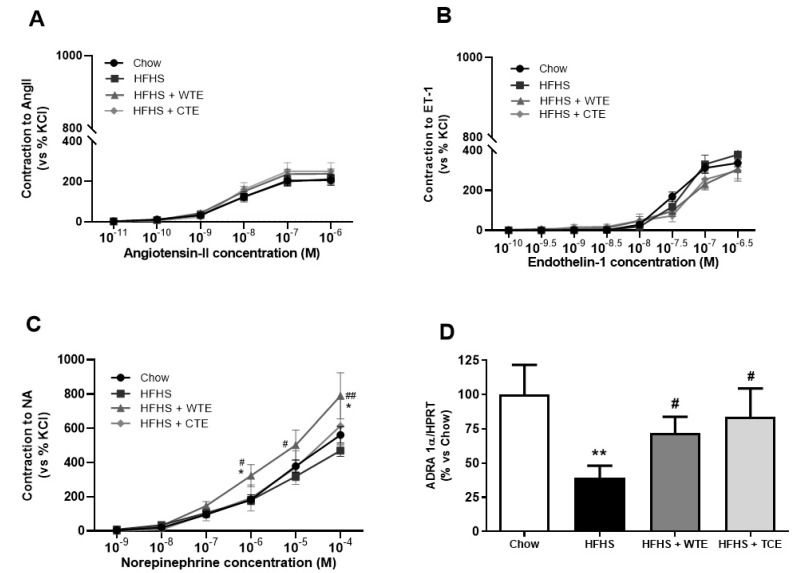
Contraction of the abdominal aortic rings to angiotensin-II (AngII) (10^−11^–10^−6^ M) (**A**), endothelin-1 (ET-1) (10^−10^–10^−6,5^ M) (**B**), noradrenaline (NA) (10^−9^–10^−4^ M) (**C**), and the gene expression of alpha-1-adrenergic receptor (ADRA 1α) (**D**) of mice fed a standard diet (Chow), a high-fat diet/sucrose diet (HFHS), a high-fat diet/sucrose diet supplemented with White Tea Extract (HFHS + WTE), or a high-fat diet/sucrose diet supplemented with Tea Complex Extract (HFHS + TCE). * *p* < 0.05; ** *p* < 0.01 vs. chow. # *p* < 0.05; ## *p* < < 0.01 vs. HFHS. The values are represented as the mean ± S.E.M; n = 7–10 samples/experimental group.

**Figure 7 antioxidants-11-01573-f007:**
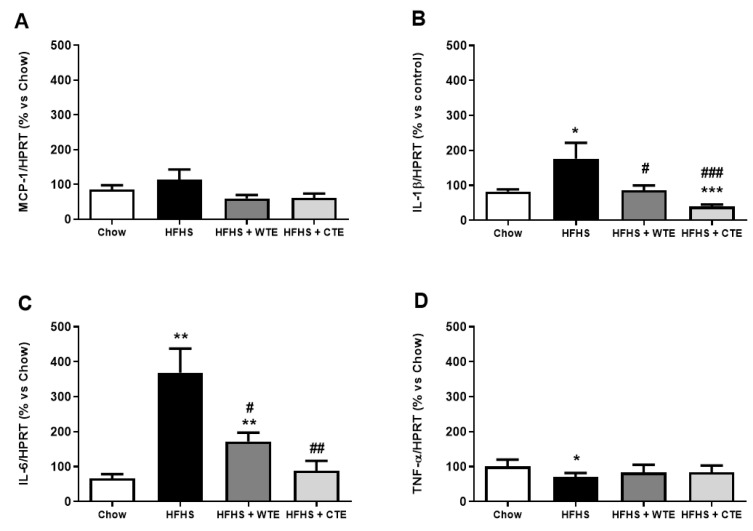
Gene expression of monocyte chemoattractant protein (MCP-1) (**A**), interleukin-1 beta (IL-1β) (**B**), interleukin-6 (IL-6) (**C**), and tumor necrosis factor-alpha (TNF-α) (**D**) of mice fed a standard diet (Chow), a high-fat diet/sucrose diet (HFHS), a high-fat diet/sucrose diet supplemented with White Tea Extract (HFHS + WTE), or a high-fat diet/sucrose diet supplemented with Tea Complex Extract (HFHS + TCE). * *p* < 0.05; ** *p* < 0.01; *** *p* < 0.001 vs. chow. # *p* < 0.05; ## *p* < 0.01 vs. HFHS. The values are represented as the mean ± S.E.M (n = 7–10 samples/experimental group), and are expressed as a percentage vs. chow.

**Figure 8 antioxidants-11-01573-f008:**
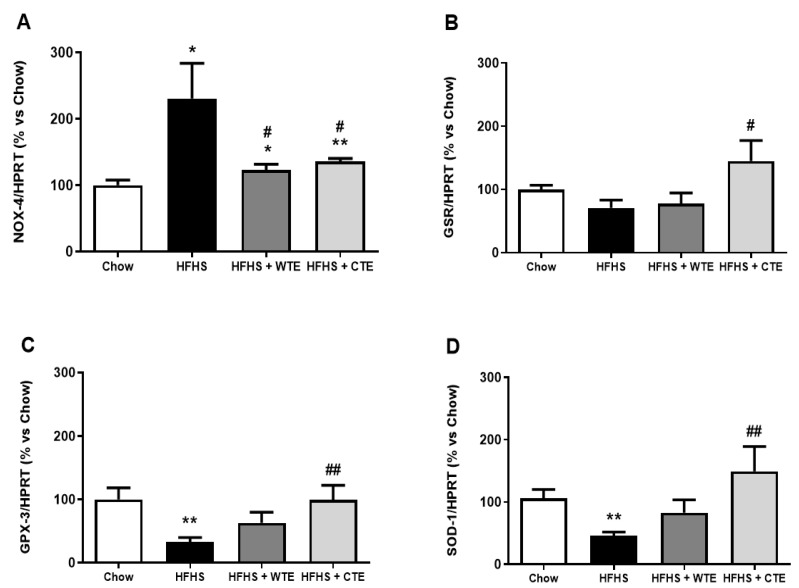
Gene expression of NADPH oxidase-4 (NOX-4) (**A**), glutathione reductase (GSR) (**B**), glutathione peroxidase 3 (GPX_3_) (**C**), and superoxide dismutase-1 (SOD-1) (**D**) of mice fed a standard diet (Chow), a high-fat diet/sucrose diet (HFHS), high-fat diet/sucrose diet supplemented with White Tea Extract (HFHS + WTE), or high-fat diet/sucrose diet supplemented with Tea Complex Extract (HFHS + TCE). * *p* < 0.05; ** *p* < 0.01 vs. chow. # *p* < 0.05; ## *p* < 0.01 vs. HFHS. The values are represented as the mean ± S.E.M (n = 7-10 samples/experimental group), and are expressed as a percentage vs. chow.

**Figure 9 antioxidants-11-01573-f009:**
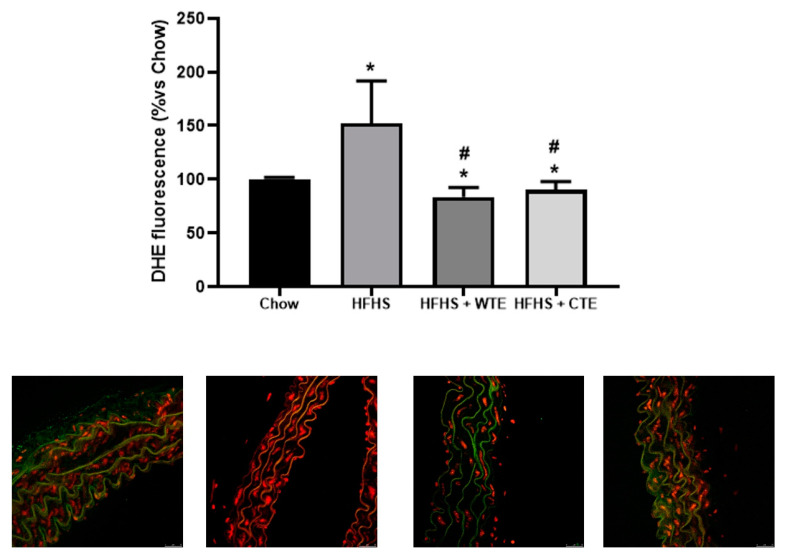
Superoxide anion generation measured as the total DHE fluorescence and respresentative fluorescent microphotographs and quantitative analysis of confocal microspy images in aortic tissue of mice fed a standard diet (Chow), a high-fat diet/sucrose diet (HFHS), a high-fat diet/sucrose diet supplemented with White Tea Extract (HFHS + WTE), or a -high-fat diet/sucrose diet supplemented with Tea Complex Extract (HFHS + TCE). * *p* < 0.05 vs. chow. # *p* < 0.05 vs. HFHS. The values are represented as the mean ± S.E.M; n = 7–10 samples/experimental group.

**Table 1 antioxidants-11-01573-t001:** Body weight (g), glycaemia (mg/dl), circulating levels of triglycerides (mg/dL), total cholesterol, LDL-cholesterol (LDL-c) and HDL cholesterol (HDL-c). The values are represented as the mean ± S.E.M (n = 7–10 samples/experimental group). *** *p* < 0.001 vs. Chow; # *p* < 0.05 vs. HFHS.

	Chow	HFHS	HFHS + WTE	HFHS + CTE
**Body weight (g)**	28.5 ± 0.5	48.7 ± 1 ***	47.7 ± 1.1	44.8 ± 0.9 ***#
**Glycaemia**	103 ± 4.9	138 ± 5.5 ***	153 ± 7 ***	142 ± 4.7 ***
**Triglycerides**	47 ± 4	61 ± 3 ***	92 ± 10 ***#	67 ± 10 ***
**Total Cholesterol**	104 ± 3	206 ± 7.4 ***	247 ± 11 ***#	238 ± 17 ***
**LDL-c**	18.8 ± 1.6	52 ± 4 ***	52.3 ± 4.6 ***	55.4 ± 6 ***
**HDL-c**	25.7 ± 1.1	52.2 ± 2.6 ***	60 ± 2.7 ***	53.7 ± 5.1 ***

## Data Availability

Data is contained within the article.

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
