# Peer review of "Supplementation with Two New Standardized Tea Extracts Prevents the Development of Hypertension in Mice with Metabolic Syndrome"

_antioxidants, 2022, doi:10.3390/antiox11081573_

Round 1

Reviewer 1 Report

The paper of De la Fuente Muñoz and colleagues assess the possible beneficial effects 20 of two new standardized tea extracts, one of white tea and the other of black and green tea on the cardiovascular alterations associated with Metabolic syndrome. The main results of paper indicate that tea extract attenuates endothelial dysfunction and prevents the development of hypertension in mice with metabolic syndrome by  decreasing arterial inflammation and oxidative stress.

The paper present different criticisms:

- The animals were indicated as a model of metabolic syndrome but niether data about the body weight, glicemia and lipid profile are presented. The authors showed only the results about the blood pressure, that is only one of the factors useful to define the metabolic sysndrome conditions. Data about other blood paremeters must be added to define the animals as a model of metabolic syndrome.

- I suggest to add data about the increase of the oxidative stress in blood plasma, and the possible antioxidants effects of the two tea extracts.

- Data about the possible cardiac alterations such as cardiomyocyte’s hypertrophy, fibrosis, coronary arteries alterations can be added.

-information about the ethical committee for the use of animal must be added. In which way was defined the sample size for the four experimental groups?

-In the material and methods is not clear in which way the two Tea extract were supplemented in the diet of animals.

- Why the dose of 1,6% for the two tea extracts?

-The quality of graphs must be improved. If is possible, the Y-axis  for the different graphs in the same pictures must be the same

Different typos are presents. For example, acetylcholine is ACh no Ach; please check the last sentence of the discussion. 

Author Response

The paper of De la Fuente Muñoz and colleagues assess the possible beneficial effects 20 of two new standardized tea extracts, one of white tea and the other of black and green tea on the cardiovascular alterations associated with Metabolic syndrome. The main results of paper indicate that tea extract attenuates endothelial dysfunction and prevents the development of hypertension in mice with metabolic syndrome by  decreasing arterial inflammation and oxidative stress.

The paper present different criticisms:

- The animals were indicated as a model of metabolic syndrome but niether data about the body weight, glicemia and lipid profile are presented. The authors showed only the results about the blood pressure, that is only one of the factors useful to define the metabolic sysndrome conditions. Data about other blood paremeters must be added to define the animals as a model of metabolic syndrome.

As suggested by the reviewer, a new table (Table 1) with data of body weight, glycaemia and lipid profile has been incorporated into de manuscript.

Chow

HFHS

HFHS+WTE

HFHS+CTE

Body weight (g)

28,5 ± 0,5

48,7 ± 1***

47,7 ± 1,1

44,8 ± 0,9***#

Glycaemia

103 ± 4,9

138 ± 5,5***

153 ± 7***

142 ± 4,7***

Triglycerides

47 ± 4

61 ± 3***

92 ± 10***

67 ± 10***

Total Cholesterol

104 ± 3

206 ± 7,4***

247 ± 11***

238 ± 17***

LDL-c

18,8 ± 1,6

52 ± 4***

52,3 ± 4,6***

55,4 ± 6***

HDL-c

25,7 ± 1,1

52,2 ± 2,6***

60 ± 2,7***

53,7 ± 5,1***

Table 1. Body weight (g), glycaemia (mg/dl) and circulating levels of triglycerides (mg/dl), total cholesterol, LDL-cholesterol (LDL-c) and HDL cholesterol (HDL-c).

These new results have been incorporated into the results section with the following paragraph: Lines 271-281

“The consumption of the HFHS diet induced a significant increase in body weight glycaemia and the plasma levels of triglycerides, total cholesterol, LDL-c and HDL-c (Table1; p<0,001 for all). None of these parameters were modified by the supplementation with the tea extracts, except for body weight that was significantly reduced in obese mice supplemented with CTE (p<0,05).”

Moreover the following paragraph has been inserted in the Discussion section (LineS 428-435)

“As expected, the consumption of the HFHS diet increased body weight, glycaemia and the circulating levels of triglycerides, total cholesterol, LDL-c and HDL-c. Neither supplementation with WTE nor with CTE had a positive impact reducing these altera-tions, except for body weight that was significantly decreased in obese mice supple-mented with CTE. The weight reduction effect of green tea has been already reported both in experimental animals [Razavi et. al 2017] and in humans [Basu et. al 2010] with MetS and seems to be due to both, decreased lipid and protein absorption in the intestine, and to the activation of the AMP- kinase pathway in the liver, skeletal muscle, and adipose tissues [Yang et. al 2016].

References:

  • Razavi, B.M.; Lookian, F.; Hosseinzadeh, H. Protective effects of green tea on olanzapine-induced-metabolic syndrome in rats. Biomed Pharmacother 2017, 92, 726-731, doi:10.1016/j.biopha.2017.05.113.
  • Basu, A.; Sanchez, K.; Leyva, M.J.; Wu, M.; Betts, N.M.; Aston, C.E.; Lyons, T.J. Green tea supplementation affects body weight, lipids, and lipid peroxidation in obese subjects with metabolic syndrome. J Am Coll Nutr 2010, 29, 31-40, doi:10.1080/07315724.2010.10719814.
  • Yang, C.S.; Zhang, J.; Zhang, L.; Huang, J.; Wang, Y. Mechanisms of body weight reduction and metabolic syndrome alleviation by tea. Mol Nutr Food Res 2016, 60, 160-174, doi:10.1002/mnfr.201500428.

- I suggest to add data about the increase of the oxidative stress in blood plasma, and the possible antioxidants effects of the two tea extracts.

We thank the reviewer for this interesting suggestion. However, that would require to use new animals because unfortunately we have run out of plasma from the mice of this animal model. According to the principle of the 3Rs (Replacement, Reduction and Refinement) we think that it is not sufficiently justified.

- Data about the possible cardiac alterations such as cardiomyocyte’s hypertrophy, fibrosis, coronary arteries alterations can be added.

We agree with the reviewer that these data would be interesting. However, we also think that they are not essential for this work since the cardiac damage associated with metabolic syndrome has been extensively reported by other authors and, above all, since supplementation with the tea extracts did not produce significant effects on cardiac function as shown in Figure 2.

-information about the ethical committee for the use of animal must be added. In which way was defined the sample size for the four experimental groups?

As suggested by the reviewer the following paragraph has been inserted in the material and methods section (Lines 150-152).

“All the experiments were conducted according to the European Union Legislation and with the approval of the Animal Care and Ethical Committee of the Community of Madrid (Spain) (PROEX 214.1_20).”

Sample size (n=7-10) was determined based on the variability observed in previous studies in order to obtain statistical differences with a p vale of at least p<0,05 among experimental groups.

-In the material and methods is not clear in which way the two Tea extract were supplemented in the diet of animals.

Customized diets were elaborated by the company Research Diets Inc. (New Brunswick, NJ, USA). The tea extracts (1,6%) were added to the commercial high fat/ high sucrose diet with reference D12331 (https://researchdiets.com/formulas/d12331). The caloric content of the tea extracts was taken into consideration in order for the diets to be isocaloric.

The diet with reference 11112201 (D11112201) was used as standard diet (chow).

This information has been inserted in the material and methos section (Lines 161-166)

- Why the dose of 1,6% for the two tea extracts?

The dose was chosen according to previous studies that have reported that supplementation with dosages between 0,5% and 5% of different types of tea extracts, especially green tea, exert anti-obesity effects in mice with metabolic syndrome. Based on these studies we decided to choose an intermediate dose (1,6%) for both extracts in order for the results to be comparable.

To make this clearer, the following paragraph has been inserted in the material and methods section. In addition a sentences regarding the equivalence between humans and rodent dosages has also been added (Lines 167-171):

The dosage of tea extracts was chosen according to previous studies in which doses between 0,5 and 5% are shown to exert beneficial effects on body weight reduction and metabolism in rodents with MetS (Bruno et. al 2008, Liu J et. al 2019, Zhang et. al 2020). Studies in humans used dosages between 10-20 times higher than the one used in this study (Chen et. al 2015, Basu et. al 2010, Basu et. al 2011), which is in the accepted range when dosages for rodents are converted into dosages for humans (Nair et. al 2016)

References:

  • Richard S. Bruno, Christine E. Dugan, Joan A. Smyth, Dana A. DiNatale, Sung I. Koo. Supplementation with a dose of 1% and 2% of green tea extract exerted positive effects of hepatic lipid deposition in leptin deficient mice. The Journal of Nutrition, Volume 138, Issue 2, February 2008, Pages 323–331. https://doi.org/10.1093/jn/138.2.323
  • Na Xu, Jun Chu, Min Wang, Ling Chen, Liang Zhang, Zhongwen Xie, Jinsong Zhang, Chi-Tang Ho, Daxiang Li, and Xiaochun Wan. Journal of Agricultural and Food Chemistry 2018 66 (15), 3823-3832. DOI: 10.1021/acs.jafc.8b00138
  • Hea Jin Park, Dana A. DiNatale, Min-Yu Chung, Young-Ki Park, Ji-Young Lee, Sung I. Koo, Meeghan O'Connor, Jose E. Manautou, Richard S. Bruno. Green tea extract attenuates hepatic steatosis by decreasing adipose lipogenesis and enhancing hepatic antioxidant defenses in ob/ob mice. The Journal of Nutritional Biochemistry. Volume 22, Issue 4, 2011, Pages 393-400, ISSN 0955-2863, https://doi.org/10.1016/j.jnutbio.2010.03.009.
  • Xiao Rong Yang, Elaine Wat, Yan Ping Wang, Chun Hay Ko, Chi Man Koon, Wing Sum Siu, Si Gao, David Wing Shing Cheung, Clara Bik San Lau, Chuang Xing Ye, Ping Chung Leung, "Effect of Dietary Cocoa Tea (Camellia ptilophylla) Supplementation on High-Fat Diet-Induced Obesity, Hepatic Steatosis, and Hyperlipidemia in Mice", Evidence-Based Complementary and Alternative Medicine, vol. 2013, Article ID 783860, 11 pages, 2013. https://doi.org/10.1155/2013/783860
  • Min-Yu Chung, Hea Jin Park, Jose E. Manautou, Sung I. Koo, Richard S. Bruno. Green tea extract protects against nonalcoholic steatohepatitis in ob/ob mice by decreasing oxidative and nitrative stress responses induced by proinflammatory enzymes. The Journal of Nutritional Biochemistry. Volume 23, Issue 4. 2012, Pages 361-367, ISSN 0955-2863, https://doi.org/10.1016/j.jnutbio.2011.01.001.
  • Sudathip Sae-tan,Connie J. Rogers,Joshua D. Lambert. Voluntary exercise and green tea enhance the expression of genes related to energy utilization and attenuate metabolic syndrome in high fat fed mice. Volume58, Issue5.10.1002/mnfr.201300621.
  • Liu J , Hao W , He Z , Kwek E , Zhao Y , Zhu H , Liang N , Ma KY , Lei L , He WS , Chen ZY. Beneficial effects of tea water extracts on the body weight and gut microbiota in C57BL/6J mice fed with a high-fat diet. Food Funct. 2019 May 22;10(5):2847-2860. doi: 10.1039/c8fo02051e.
  • Zhang Y, Gu M, Wang R, Li M, Li D, Xie Z. Dietary supplement of Yunkang 10 green tea and treadmill exercise ameliorate high fat diet induced metabolic syndrome of C57BL/6 J mice. Nutr Metab (Lond). 2020 Feb 4;17:14. doi: 10.1186/s12986-020-0433-9. eCollection 2020.
  • Liu Z, Xiao M, Du Z, Li M, Guo H, Yao M, Wan X, Xie Z. Dietary supplementation of Huangshan Maofeng green tea preventing hypertension of older C57BL/6 mice induced by desoxycorticosterone acetate and salt. J Nutr Biochem. 2021 Feb;88:108530. doi: 10.1016/j.jnutbio.2020.108530.
  • Chen IJ, Liu CY, Chiu JP, Hsu CH. Therapeutic effect of high-dose green tea extract on weight reduction: A randomized, double-blind, placebo-controlled clinical trial. Clin Nutr. 2016 Jun;35(3):592-9. doi: 10.1016/j.clnu.2015.05.003. Epub 2015 May 29. PMID: 26093535.
  • Basu A, Sanchez K, Leyva MJ, Wu M, Betts NM, Aston CE, Lyons TJ. Green tea supplementation affects body weight, lipids, and lipid peroxidation in obese subjects with metabolic syndrome. J Am Coll Nutr. 2010 Feb;29(1):31-40. doi: 10.1080/07315724.2010.10719814. PMID: 20595643.
  • Basu A, Du M, Sanchez K, Leyva MJ, Betts NM, Blevins S, Wu M, Aston CE, Lyons TJ. Green tea minimally affects biomarkers of inflammation in obese subjects with metabolic syndrome. Nutrition. 2011 Feb;27(2):206-13. doi: 10.1016/j.nut.2010.01.015. Epub 2010 Jun 2. PMID: 20605696; PMCID: PMC2952043.
  • Nair AB, Jacob S. A simple practice guide for dose conversion between animals and human. J Basic Clin Pharm. 2016 Mar;7(2):27-31. doi: 10.4103/0976-0105.177703. PMID: 27057123; PMCID: PMC4804402.

-The quality of graphs must be improved. If is possible, the Y-axis  for the different graphs in the same pictures must be the same

It has been corrected.

Different typos are presents. For example, acetylcholine is ACh no Ach; please check the last sentence of the discussion.

The manuscript has been carefully revised and all the detected typo errors have been corrected.

Reviewer 2 Report

The manuscript entitled “Supplementation with two new standardized tea extracts prevents the development of hypertension in mice with metabolic syndrome” described the preventive effect of white tea extracts (WTE) and black and green tea extracts (CTE) on the development of hypertension in mice fed with high fat and high sugar diet. However, there are some concerns for this manuscript. In my opinion, the manuscript could be accepted for publication after a major revision.

Major points

1.      The authors should fully explain the necessity of studying white tea, especially the mixture of black and green tea. The present study revealed that WTE and CTE attenuates endothelial dysfunction and prevents the development of hypertension in mice, but the comparative effect of WTE and XTE was not discussed enough. The authors should explain it more.

2.      The authors should state the dose basis of WTE and CTE in animal experiment. Are the doses achievable in humans by dietary intake?

3.      The authors claimed that WTE and CTE attenuates endothelial dysfunction and prevents the development of hypertension in mice with metabolic syndrome by decreasing arterial inflammation and oxidative stress. However, the present study only evaluated the proinflammatory cytokines and oxidative stress related markers on transcription level. The authors should also measure the protein levels of them.

4.      The underlying molecular mechanisms of antihypertensive actions of WTE and CTE are limited in this study.

Minor points

1. Line 403. “the hepatotoxic effects Of” should be changed into “the hepatotoxic effects of”.

2. Line 485-486, the conclusions section should be deleted.

Author Response

The manuscript entitled “Supplementation with two new standardized tea extracts prevents the development of hypertension in mice with metabolic syndrome” described the preventive effect of white tea extracts (WTE) and black and green tea extracts (CTE) on the development of hypertension in mice fed with high fat and high sugar diet. However, there are some concerns for this manuscript. In my opinion, the manuscript could be accepted for publication after a major revision.

Major points

  1. The authors should fully explain the necessity of studying white tea, especially the mixture of black and green tea. The present study revealed that WTE and CTE attenuates endothelial dysfunction and prevents the development of hypertension in mice, but the comparative effect of WTE and XTE was not discussed enough. The authors should explain it more.

As suggested by the reviewer, the following paragraph about the comparative effect between WTE and CTE has been inserted in the Discussion section: (Lines 500-507)

An important finding of this work is that the tea extracts tested in this study not only decrease oxidative stress but also inflammation in arterial tissue, an issue that may contribute to their antihypertensive effects. In this regard, and contrary to the antioxi-dant effect that is similar after the supplementation with both extracts, our results show that CTE exerts a more powerful anti-inflammatory effect in arterial tissue than WTE. This higher anti-inflammatory effect maybe due, at least in part, to the different composition of the extracts, and particularly to the higher amount of gallic acid, xanthines and fla-van-3-ols in CTE compared to WTE. Indeed, the anti-inflammatory effects of gallic acid (Bai et. al 2021) , xanthines (Sing et. al 2018) and flavan-3-ols (Mena et. al 2014) are extensively reported and may contribute to the more pronounced antihypertensive effect of CTE compared to WTE.

References:

  • Bai, J.; Zhang, Y.; Tang, C.; Hou, Y.; Ai, X.; Chen, X.; Zhang, Y.; Wang, X.; Meng, X. Gallic acid: Pharmacological activities and molecular mechanisms involved in inflammation-related diseases. Biomed Pharmacother 2021, 133, 110985, doi:10.1016/j.biopha.2020.110985.
  • Singh, N.; Shreshtha, A.K.; Thakur, M.S.; Patra, S. Xanthine scaffold: scope and potential in drug development. Heliyon 2018, 4, e00829, doi:10.1016/j.heliyon.2018.e00829.
  • Mena, P.; Dominguez-Perles, R.; Girones-Vilaplana, A.; Baenas, N.; Garcia-Viguera, C.; Villano, D. Flavan-3-ols, anthocyanins, and inflammation. IUBMB Life 2014, 66, 745-758, doi:10.1002/iub.1332.
  1. The authors should state the dose basis of WTE and CTE in animal experiment. Are the doses achievable in humans by dietary intake?

In this study we have administered a dosage of 1,6% of WTE and CTE in the chow. Taking into account that each mouse eat approximately 3g of chow per day, the daily amount of tea extract per mouse is 48 mg/day. This amount is comparable to the one used by other authors in experimental models of metabolic syndrome. In most of the animal studies this dose ranges between 0,5 and 5%. For example, in the study by Liu J et. al a dose of 1% of tea extract was effective to improve glucose tolerance and to reduce gain in weight, hepatic lipids, and white adipose tissue weights in mice with metabolic syndrome. In the study performed by Zhang et. al., a higher dose (5%) exerted similar effects. Thus, we preferred to use a dose closer to 1% but a little bit higher, to make sure that in addition to the metabolic effects, cardiovascular effects will be also present. In this regard, our tea extracts are more effective preventing the development of hypertension than other tea extracts that have been tested at higher dosages (2,5% and 5%) (Liu Z et. al 2021).

For humans,  the recommended daily intake of green tea for humans is 3 to 5 cups/day (720 to 1,200 mL) which provides around 180 mg of catechins and at least 60 mg of theanine.

In general, the dosages of botanic extracts administered to experimental animals are usually much higher than dosages in humans, in a range of 10 to 30 times higher (https://www.targetmol.com/pages/dosage).

Regarding tea extracts, clinical studies reported in the literature have used different dosages with positive effects on metabolism and weight reduction. For example, a study performed in women with central obesity a dose of 856,8mg/day green tea extract was used (Chen et. al 2015). This dose was 18 times higher than the one used in our study. However, a dose of 500mg/day of green tea, even though it reduced BMI and body weight (Basu et. al 2010a), it did not show positive effects on the expression of circulating inflammatory markers in obese individuals with MetS (Basu et. al 2011). Thus, it is possible that in humans a dose higher than 500mg/dat is necessary to improve metabolic and cardiovascular function in patients with MetS.

To make this issue clearer, the following paragraph has been inserted in the material and methods section: Lines 167-171

The dosage of tea extracts was chosen according to previous studies in which doses between 0,5 and 5% are shown to exert beneficial effects on body weight reduction and metabolism in rodents with MetS (Bruno et. al 2008, Liu J et. al 2019, Zhang et. al 2020). Studies in humans used dosages between 10-20 times higher than the one used in this study (Chen et. al 2015, Basu et. al 2010, Basu et. al 2011), which is in the accepted range when dosages for rodents are converted into dosages for humans (Nair et. al 2016)

References

Richard S. Bruno, Christine E. Dugan, Joan A. Smyth, Dana A. DiNatale, Sung I. Koo. Supplementation with a dose of 1% and 2% of green tea extract exerted positive effects of hepatic lipid deposition in leptin deficient mice. The Journal of Nutrition, Volume 138, Issue 2, February 2008, Pages 323–331. https://doi.org/10.1093/jn/138.2.323

Liu J , Hao W , He Z , Kwek E , Zhao Y , Zhu H , Liang N , Ma KY , Lei L , He WS , Chen ZY. Beneficial effects of tea water extracts on the body weight and gut microbiota in C57BL/6J mice fed with a high-fat diet. Food Funct. 2019 May 22;10(5):2847-2860. doi: 10.1039/c8fo02051e.

Zhang Y, Gu M, Wang R, Li M, Li D, Xie Z. Dietary supplement of Yunkang 10 green tea and treadmill exercise ameliorate high fat diet induced metabolic syndrome of C57BL/6 J mice. Nutr Metab (Lond). 2020 Feb 4;17:14. doi: 10.1186/s12986-020-0433-9. eCollection 2020.

Liu Z, Xiao M, Du Z, Li M, Guo H, Yao M, Wan X, Xie Z. Dietary supplementation of Huangshan Maofeng green tea preventing hypertension of older C57BL/6 mice induced by desoxycorticosterone acetate and salt. J Nutr Biochem. 2021 Feb;88:108530. doi: 10.1016/j.jnutbio.2020.108530.

Chen IJ, Liu CY, Chiu JP, Hsu CH. Therapeutic effect of high-dose green tea extract on weight reduction: A randomized, double-blind, placebo-controlled clinical trial. Clin Nutr. 2016 Jun;35(3):592-9. doi: 10.1016/j.clnu.2015.05.003. Epub 2015 May 29. PMID: 26093535.

Basu A, Sanchez K, Leyva MJ, Wu M, Betts NM, Aston CE, Lyons TJ. Green tea supplementation affects body weight, lipids, and lipid peroxidation in obese subjects with metabolic syndrome. J Am Coll Nutr. 2010 Feb;29(1):31-40. doi: 10.1080/07315724.2010.10719814. PMID: 20595643.

Basu A, Du M, Sanchez K, Leyva MJ, Betts NM, Blevins S, Wu M, Aston CE, Lyons TJ. Green tea minimally affects biomarkers of inflammation in obese subjects with metabolic syndrome. Nutrition. 2011 Feb;27(2):206-13. doi: 10.1016/j.nut.2010.01.015. Epub 2010 Jun 2. PMID: 20605696; PMCID: PMC2952043.

Nair AB, Jacob S. A simple practice guide for dose conversion between animals and human. J Basic Clin Pharm. 2016 Mar;7(2):27-31. doi: 10.4103/0976-0105.177703. PMID: 27057123; PMCID: PMC4804402.

  1. The authors claimed that WTE and CTE attenuates endothelial dysfunction and prevents the development of hypertension in mice with metabolic syndrome by decreasing arterial inflammation and oxidative stress. However, the present study only evaluated the proinflammatory cytokines and oxidative stress related markers on transcription level. The authors should also measure the protein levels of them.

We agree with the reviewer that it would be advisable to also measure the protein levels of the inflammatory and oxidative stress related markers. However, that would require to use new animals because unfortunately we have run out of arterial tissue from the mice of this animal model. Following the principle of the 3Rs (Replacement, Reduction and Refinement) we don’t think that it is enough justified.

  1. The underlying molecular mechanisms of antihypertensive actions of WTE and CTE are limited in this study.

In this study we report that supplementation with WTE or CTE prevents the development of hypertension in mice with metabolic syndrome through improvement of endothelial function. Moreover, we demonstrate that this improvement is mediated by the antioxidant and anti-inflammatory effects of the tea extracts. The antioxidant effect is demonstrated not only by the measurement of the mRNA levels of antioxidant enzymes in arterial tissue but also through the quantification of superoxide anion in the aorta. In this regard, our results show that supplementation with the tea extracts totally prevent the obesity-induced increase in ROS content in the aorta which, in our opinion, gives a very certain idea of the possible underlying mechanism of the antihypertensive effects.  

Minor points

  1. Line 403. “the hepatotoxic effects Of” should be changed into “the hepatotoxic effects of”.

It has been corrected.

  1. Line 485-486, the conclusions section should be deleted.ç

Following the indications of the Editor conclusions are now in a separate section (section 5) of the manuscript.

Round 2

Reviewer 1 Report

The authors reply point by point at all request. 

Paper can be accepted in this form

Author Response

We thank the reviewer for the acceptance of the mansucript

Reviewer 2 Report

Authors have addressed some of the questions and concerns in the revised version. However, authors failed to address the comments requiring experimental data in support. Hence, I do not recommend this manuscript for publication in Antioxidants since the quality presented does not reach the journal's standards. 

Author Response

We are puzzled and disappointed about this unexpected decision by reviewer 2. We do not understand that, after having responded to most of the comments and suggestions in the first review, the evaluation of the manuscript is now much lower compared to the first submitted version. In our opinion, and also based on the opinion of Reviewer 1 the manuscript has been substantially improved compared to the first version.

On the other hand, it is not clear to us what the reviewer means by “authors failed to address the comments requiring experimental data in support.” We haven’t answered to this comment because, to our knowledge, this comment did not appear in his/her first review.
